# Molecular phylogeny and species delimitation of the genus *Tonkinacris* (Orthoptera, Acrididae, Melanoplinae) from China

Haojie Wang [1☉], Bing Jiang [2,3☉], Jingxiao Gu [2,3], Tao Wei [4], Liliang Lin [5], Yuan Huang [5], Dan Liang [1]*, Jianhua Huang [2,3]*

**1** College of Biological Sciences and Technology, Beijing Forestry University, Beijing, People's Republic of China, **2** Key Laboratory of Insect Evolution and Pest Management for Higher Education in Hunan Province, Central South University of Forestry and Technology, Changsha, Hunan, People's Republic of China, **3** Key Laboratory of Cultivation and Protection for Non–Wood Forest Trees (Central South University of Forestry and Technology), Ministry of Education, Changsha, Hunan, People's Republic of China, **4** Tanxi Street Agency, Liunan Subdistrict, Liuzhou, Guangxi, People's Republic of China, **5** College of Life Sciences, Shaanxi Normal University, Xi'an, Shaanxi, People's Republic of China

☉ These authors contributed equally to this work.
* liangdanyx2014@163.com (DL); caniscn@aliyun.com (JH)

**Data Availability Statement:** All new haplotype nucleotide sequences are available from the GenBank database (accession numbers are given in S3 Table).

## Abstract

*Tonkinacris* is a small group in Acrididae. While a few species were occasionally sampled in some previous molecular studies, there is no revisionary research devoted to the genus. In this study, we explored the phylogeny of and the relationships among Chinese species of the genus *Tonkinacris* using the mitochondrial *COI* barcode and the complete sequences of *ITS1* and *ITS2* of the nuclear ribosomal DNA. The phylogeny was reconstructed in maximum likelihood and Bayesian inference frameworks, respectively. The overlap range between intraspecific variation and interspecific divergence was assessed via K2P distances. Species boundaries were delimitated using phylogenetic species concept, NJ tree, K2P distance, the statistical parsimony network as well as the GMYC model. The results demonstrate that the Chinese *Tonkinacris* species is a monophyletic group and the phylogenetic relationship among them is (*T. sinensis*, (*T. meridionalis*, (*T. decoratus*, *T. damingshanus*))). While *T. sinensis*, *T. meridionalis* and *T. decoratus* were confirmed being good independent species strongly supported by both morphological and molecular evidences, the validity of *T. damingshanus* was not perfectly supported by molecular evidence in this study.

## Introduction

The genus *Tonkinacris* Carl, 1916 is a small group of grasshoppers with six known species worldwide so far [1–6]. Among the four species having distribution in China, *T. sinensis* has the most wide distributional range covering northeastern Yunnan, Sichuan, Guizhou, Chongqing, southern and western parts of Hubei and Hunan, Guangxi and northern Vietnam, *T. decoratus* is distributed mainly in south Guangxi and north Vietnam, and *T. damingshanus*

**Funding:** This study is supported by the Open foundation for innovation platform of Education Bureau of Hunan Province (18K056), Beijing Forestry University Teaching Reform Project (BJFU2017JY021) and National Natural Science Foundation of China (No. 31540055, 31260523, 31801993).

and *T. meridionalis* are both endemic to China with collection records only from the type localities to date. *T. ruficerus* and *T. yaeyamaensis* are distributed only in Ryukyu Islands, Japan (Fig 1).

Although *Tonkinacris* was involved in a few monographs on grasshopper taxonomy [7–13], and a few species of the genus were occasionally sampled in some molecular studies (S1 Table) [14–22], there was no revisionary research devoted to the genus based on morphological or molecular data. *Tonkinacris* usually had a closest relationship to the genera *Sinopodisma* [14,16–18], *Pedopodisma* [23,24], *Podisma* [15], *Fruhstorferiola* [19–21] or *Ognevia* [22] in the molecular studies above-mentioned depending on the differences in sampling strategies. However, phylogenetic relationship within the genus *Tonkinacris* has never been explored until now.

For morphological species identification, *T. sinensis* can be easily distinguished by the median black longitudinal stripe on the dorsum of pronotum narrower and lateral yellowish

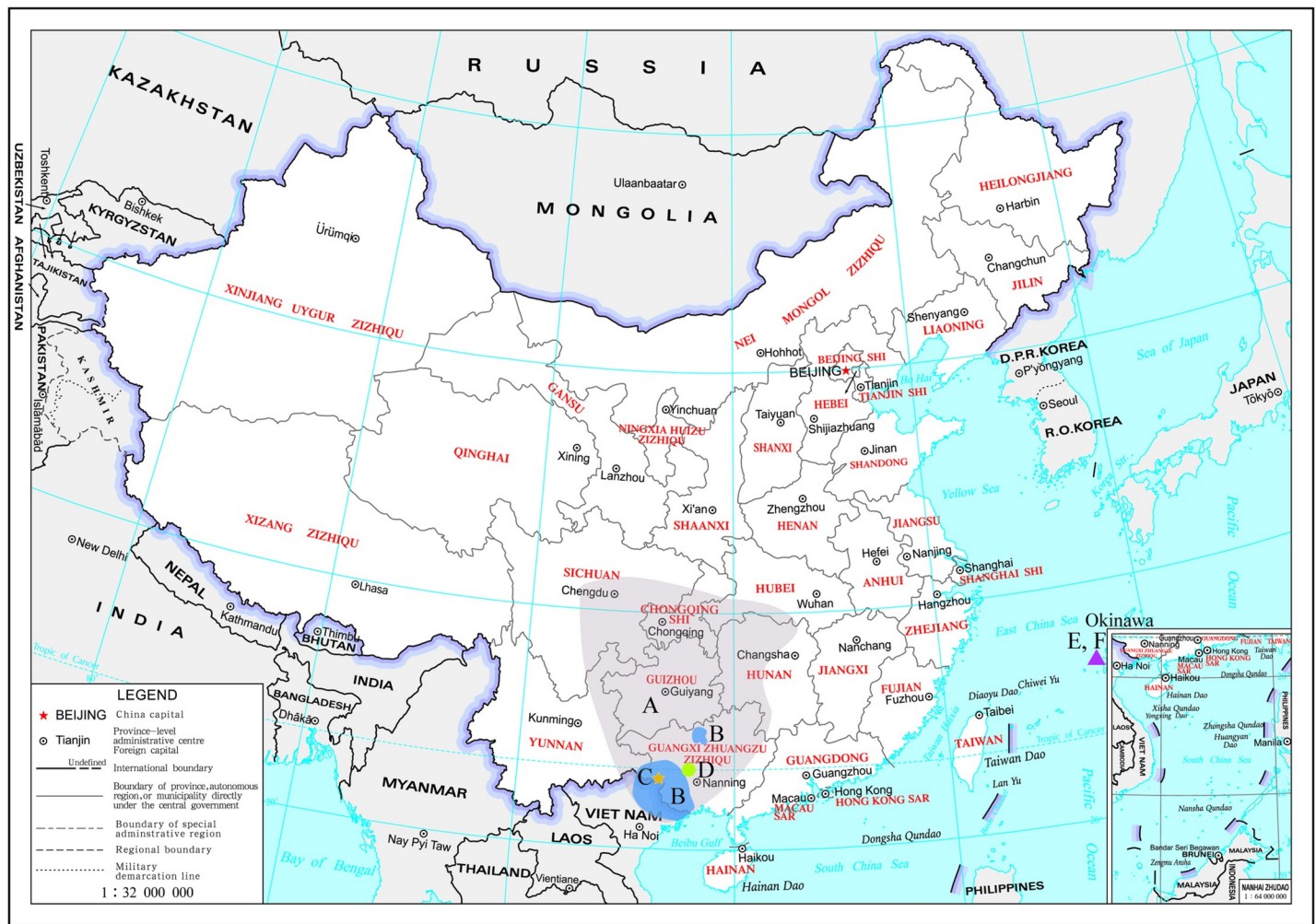

**Fig 1. Distributional ranges of *Tonkinacris spp*.** A. *T. sinensis*. B. *T. decoratus*. C. *T. meridionalis*. D. *T. damingshanus*. E, F. *T. ruficerus* and *T. yaeyamaensis* (The ground map was reprinted from Standard Map Service (http://bzdt.ch.mnr.gov.cn) under a CC BY license, with permission from the Ministry of Natural Resources of the People's Republic of China under the permission number GS(2019)1679, original copyright 2019; the ground map is available at: http://bzdt.ch.mnr.gov.cn/browse. html?picId = %224o28b0625501ad13015501ad2bfc0281%22).

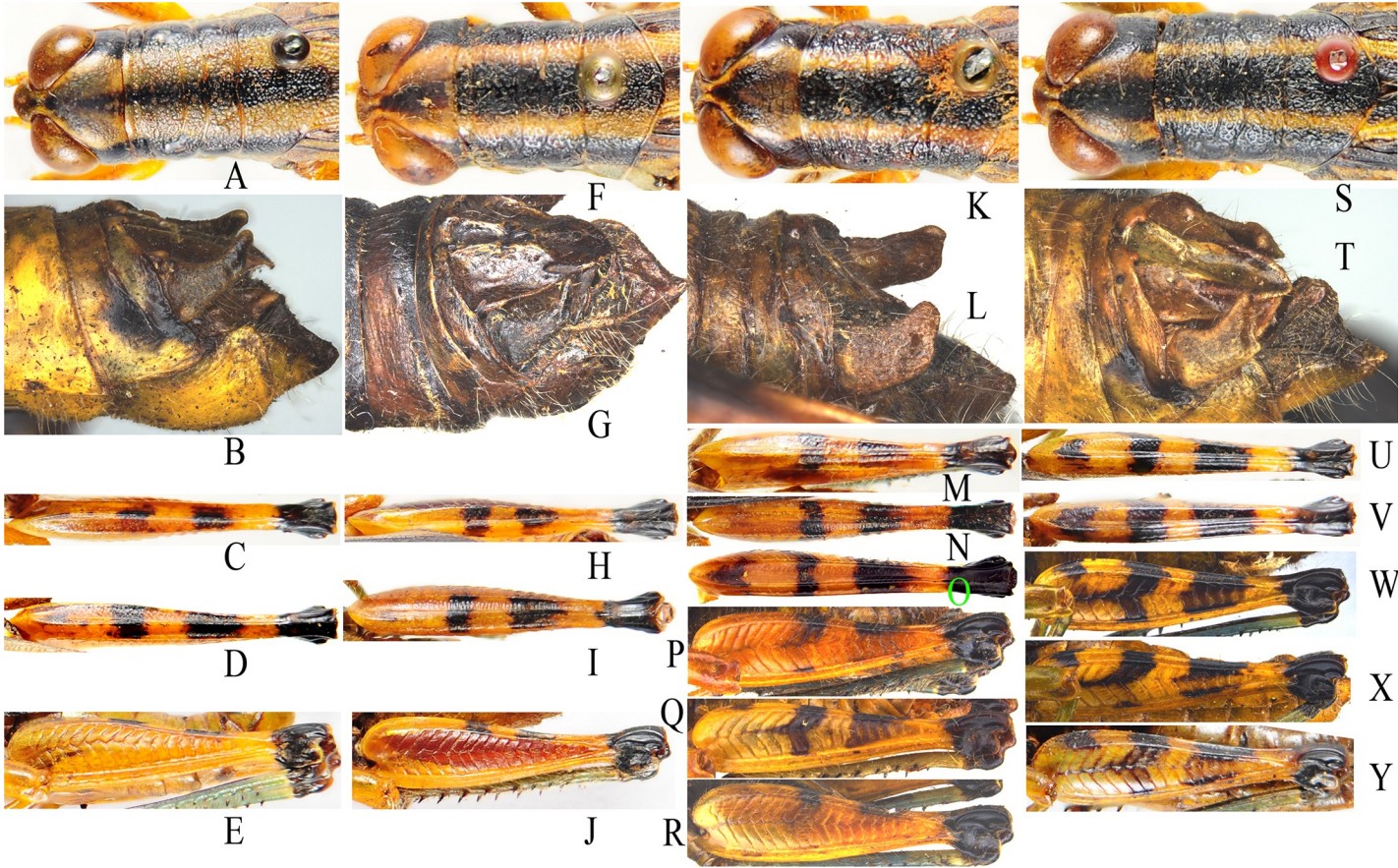

**Fig 2. Pronotum, male cerci and hind femur of *Tonkinacris spp*.** A-E. *T. sinensis*. F-J. *T. meridionalis*. K-Q. *T. decoratus*. R-X. *T. damingshanus*. A, F, K, R. Pronotum. B, G, L, S. Male cerci. C, D, H, I, M, N, T, U. Upper surface of hind femur. E, J, O–Q, V–X. External surface of hind femur.

ones broader than those in three other species, the immaculate external surfaces of hind femora and the subconical cerci in male (Fig 2A–2E). However, *T. meridionalis* seems to be an intermediate species between *T. sinensis* and the group comprising *T. decoratus* and *T. damingshanus*. That's to say, *T. meridionalis* is similar to *T. sinensis* in the immaculate external surfaces of hind femora and conical cerci in male (Fig 2G and 2J), but similar to *T. decoratus* and *T. damingshanus* in the pattern of pronotal stripes and the overall appearance of the body (Fig 2F). *T. decoratus* and *T. damingshanus* are the most similar pair of species in the genus. According to the original description [4], the minor difference between them is the three complete black transverse maculations on the upper surfaces of hind femora for *T. damingshanus* (Fig 2U and 2V). *T. decoratus* usually has only two black transverse maculations on the upper surfaces of hind femora (Fig 2M and 2N). But variation in this character was observed in *T. decoratus*, of which some individuals have also three black transverse maculations on the upper surfaces of hind femora (Fig 2O and 2P). Both *T. decoratus* and *T. damingshanus* have similar patterns of variations in the black transverse maculations on the external surfaces of hind femora (Fig 2P–2R and 2W–2Y). It seems that the morphological difference between them nearly disappears and the validity of *T. damingshanus* becomes questionable.

Since molecular data has demonstrated its significant implication in exploring phylogeny and species delimitation in many grasshopper groups (see reference [22] for a short review), we decided to investigate through molecular approaches the phylogeny of Chinese *Tonkinacris*

species and to clarify the relationship between *T. decoratus* and *T. damingshanus*. The two Japanese *Tonkinacris* species were not included in this study because they are morphologically extremely different from the four species distributed in continental China and Vietnam and may not belong to the genus *Tonkinacris*. More importantly, the only mitochondrial *COI* fragments sequenced by Grzywacz *et al.* [20] for the Japanese species *T. ruficerus* and *T. yaeyamaensis* do not overlap even partially with, but are distantly separated from, the standard barcoding region sequenced in this study and our previous similar studies [18,22]. Therefore, the partial *COI* sequences of *T. ruficerus* and *T. yaeyamaensis* sequenced in Grzywacz *et al.*'s study [20] cannot be combined into our dataset for analysis. In this study, we sampled 215 individuals belonging to 15 genera and 20 species in Acrididae and sequenced the 658-base fragment of the barcode region in mitochondrial *COI* gene for animals [25], and the complete sequences of *ITS1* and *ITS2* of the nuclear ribosomal DNA for 149 individuals. Phylogeny of the species involved in this study was reconstructed from molecular sequence datasets using maximum likelihood and Bayesian inference methods, and the species boundary was delimitated for Chinese *Tonkinacris* species using multiple methods, including genetic distance, NJ tree, the haplotype network constructed using the statistical parsimony method [26] and analysis of the generalized mixed Yule coalescent model (GMYC) [27]. The results demonstrate that Chinese species of the genus *Tonkinacris* is a monophyletic group and the phylogenetic relationship among them is (*T. sinensis*, (*T. meridionalis*, (*T. decoratus*, *T. damingshanus*))). While *T. sinensis*, *T. meridionalis* and *T. decoratus* were confirmed being good independent species strongly supported by both morphological and molecular evidences, the validity of *T. damingshanus* was not perfectly supported by molecular evidence in this study.

## Materials and methods

### Taxon sampling and generation of molecular sequences

A total of 218 individuals representing 2 suborders 3 families 17 genera and 22 species were sampled (S2 Table). The sampling strategy was the same as that in our previous studies [18,22]. Species assignation of specimens was performed mainly following Li & Xia's [11] keys to species. All specimens were preserved in anhydrous ethanol and stored at room temperature. The protocols for DNA extraction, PCR amplification, sequencing, sequence assembly and alignment followed those in our previous studies [18,22]. *COI* barcode region for animals, *ITS1* and *ITS2* were newly sequenced for 149 individuals and the remaining sequences were derived from our previous studies (S3 Table) [18,22]. Haplotype nucleotide sequences were deposited in GenBank with accession numbers MW053510–MW053544, MW056459–MW056489 for *COI* haplotypes, and MW054567–MW054626, MW055691–MW055701 for *ITS* sequences. *T. ruficerus* and *T. yaeyamaensis* were not included in the analysis because the fragment sequenced by Grzywacz & Tatsuta [20] is not the barcode region. The twenty samples of *T. decoratus* were collected from three localities in Nonggang Nature Reserve, Longzhou County, Guangxi, with gh050 (Fig 2O) having an extremely distinct black transverse maculation on the base of the upper surface of hind femur, gh063 having a slightly distinct black transverse maculation, and gh054, gh062, gh067 as well as gh069 having an indistinct black transverse maculation, respectively.

## Data analysis

Sequence divergences were calculated using the Kimura two parameter (K2P) distance model to explore the extent of overlap between intraspecific variation and interspecific divergence [28,29]. A neighbor-joining (NJ) tree of K2P distances was created to provide a graphic representation of the patterning of divergence between species [30] as a profile for the setup of taxa

and groups for calculating genetic distances and a reference framework for species delimitation. The calculation of the sequence divergences and NJ tree building with 1000 bootstrap replicates were implemented in MEGAX [31].

The phylogeny was reconstructed in maximum likelihood and Bayesian inference frameworks with *Ergatettix dorsiferus* in Tetrigidae and *Conocephalus longipennis* in Tettigoniidae as outgroups. The sequences of *COI* were divided into three subsets which consisted of the first, second or third position of the codons, respectively. Maximum-Likelihood phylogenies were reconstructed using IQ-TREE [32], best-fit models of nucleotide evolution and best-fit partitioning scheme were selected using ModelFinder [33], the approximately unbiased branch support values were calculated using UFBoot2 [34], and the analysis was performed in W-IQ-TREE [35] using default settings most of the time. BI analyses were accomplished in mrbayes 3.2.1 (http://morphbank.Ebc.uu.SE/mrbayes/) [36], with two independent runs, each with four Markov Chain Monte Carlo (MCMC) chains. The analysis was run for $1\times10^7$ generations, sampling every 100 generations, and the first 25% generations were discarded as burn-in, whereas the remaining samples were used to summarize Bayesian posterior probabilities (PP).

Considering the possible simple correspondence between the identity of traditional species or evolutionarily significant units (ESUs) and an objective standard of genetic differentiation: the 95% connection limit in statistical parsimony networks [27,37–39], we constructed haplotype networks for *Tonkinacris* species. The construction of haplotype networks was implemented in TCS1.21 [40].

Since the generalized mixed Yule coalescent model (GMYC) was considered a robust tool for delimiting species when only single-locus information was available [41], the single-threshold GMYC analysis of *COI* sequences was conducted in Rv3.6.1 in a Windows environment with the use of the *splits* package. The ultrametric single-locus gene tree required for the GMYC method was obtained using BEAST 1.8.2 [42] with $1\times10^7$ MCMC generations under the Yule speciation model and a burn-in of the first 10% generations was used to avoid suboptimal trees in the final consensus tree.

## Results

### Phylogeny

The phylogeny was reconstructed in both maximum likelihood (ML) and Bayesian inference (BI) frameworks using three different datasets, respectively.

The ML tree inferred from *COI* sequences retrieved monophyly for the genus *Tonkinacris* and all species with extremely high bootstrap values except *Fruhstorferiola tonkinensis*, of which the bootstrap value for the clade is only 55 (Fig 3). However, the monophyly of the genus *Tonkinacris* was not supported, and the sepcies *T. sinensis*, *T. meridionalis*, *Longgenacris maculacarina*, *Paratonkinacris vittifemoralis* and *Emeiacris maculata* did not individually cluster into an independent clade in the ML tree from *ITS1*sequences (S1 Fig). The situation became more rotten in the ML tree from *ITS2* sequences, and more species such as *T. decoratus*, *T. damingshanus*, *Apalacris varicornis* and *Xenocatantops brachycerus* were not resolved as monophyletic (S2 Fig). The poor performance of *ITS1* and *ITS2* sequences in resolving phylogeny directly led to a non-monophyly of the genus *Tonkinacris* and the species *T. sinensis* in the ML tree from the combined alignment of *COI*, *ITS1* and *ITS2* sequences, but all species out of the genus *Tonkinacris* were monophyletic with extremely high bootstrap values (S3 Fig). Among the four species within the genus *Tonkinacris*, *T. decoratus* and *T. damingshanus* have the closest relationship, and the relationship among the species of *Tonkinacris* can be described as (((*T. decoratus*, *T. damingshanus*), *T. meridionalis*), *T. sinensis*).

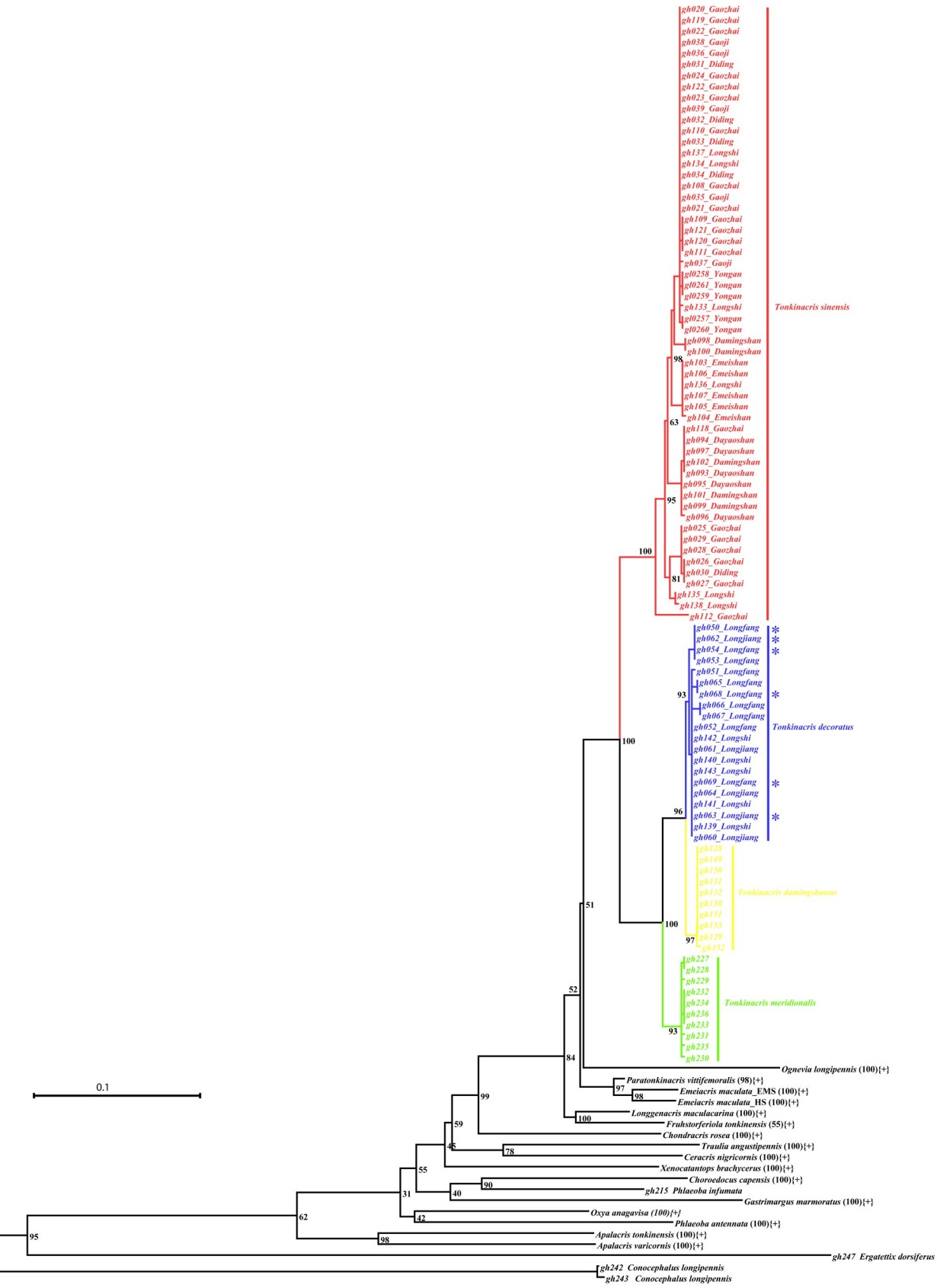

**Fig 3. ML tree deduced from mitochondrial *COI* sequences.** Subclades of species with more than one individual sampled are collapsed and marked with {+} except those of the four species of *Tonkinacris*. The number in parentheses indicates its bootstrap support value. The asterisk (*) indicates the individuals of *T. decoratus* with distinct or indistinct black transverse maculation on the base of the upper surface of hind femur.

Phylogeny revealed by BI analyses was similar to that by ML analyses. In BI trees deduced from *COI* (S4 Fig) and combined sequences (Fig 4), the monophyly of the genus *Tonkinacris* and all species was strongly supported with the clades of most species having a posterior probability value of 1 except that of *T. decoratus*, which had a relatively lower posterior probability value of 0.84 (Fig 4) and 0.93 (S4 Fig), respectively. The relationship among the four species within the genus *Tonkinacris* was the same as that revealed in ML tree by *COI* sequences. In BI tree from *ITS1* sequences (S5 Fig), the monophyly of distantly related species was strongly supported with posterior probability value of 1 except *Longgenacris maculacarina*, of which one individual fell into the clade of *T. sinensis*. For the closely related species such as the *Tonkinacris* species, *Paratonkinacris vittifemoralis* and *Emeiacris maculata*, none of them formed monophyletic group and the relationship among them was entirely unresolved. In BI tree from *ITS2* sequences (S6 Fig), while most of the distantly related species were monophyletic with posterior probability value of 1, the clade of *Apalacris tonkinensis* had only a low posterior probability value of 0.53, and *Apalacris varicornis* as well as *Xenocatantops brachycerus* were not monophyletic. The relationship among species was still entirely unresolved.

As for the 6 individuals of *T. decoratus* with distinct or indistinct black transverse maculation on the base of the upper surface of hind femur, they did not form an independent clade either by themselves or with individuals of *T. damingshanus* in both ML and BI trees (Figs 2 and 3).

## Intraspecific variation and interspecific divergence

K2P distances within and between species/populations were calculated to measure intraspecific variation and interspecific divergence, respectively. The results showed similar distribution patterns to that in our previous study (S4 Table) [18,22]. For *COI* sequences, variations within population were mostly distinctly less than or slightly above 1%, except those of *T. sinensis* within Diding, Damingshan, Nonggang and Gaozhai populations, of which the maximum pairwise distance was 1.86%, 1.86%, 1.70% and 3.13%, respectively. Mean interpopulation variations for *T. sinensis* ranged from 0.18% (between Yong'an and Gaoji populations) to 1.70% (between Dayaoshan and Emeishan populations; Table 1), and those for *Emeiacris maculata* ranged from 4.24% to 4.73% (average 4.45%). The pairwise distances more than 3.00% for *T. sinensis* within Gaozhai population and between populations were all derived from the extreme sequence gh112 (S4 Table). The pairwise distances between gh112 and other individuals of *T. sinensis* ranged from 2.33% to 3.29%. Except the distances between gh112 and gh138 as well as gh135 from Nonggang population were 2.33% and 2.49%, respectively, all other distances were more than 2.81%, and most of them were more than 2.97%. Among all intraspecific pairwise distances of *T. sinensis*, only 106 out of 1540 (representing 6.88% of the total) were more than 2.00%, and 55 were derived from gh112, 14 from gh030 of Diding population, 28 from gh026 and gh027 of Gaozhai population, respectively, and the remaining 9 from gh025, gh028, gh029 of Gaozhai population. In a word, 93.12% of the total intraspecific variations were less than 2.00%, most of the high intraspecific variations more than 2.00% were derived from gh112 and other five individuals of Gaozhai population (gh025, gh026, gh027, gh028, gh029), and only one individual (gh030 from Diding population) out of Gaozhai population had 14 intraspecific distances more than 2.00%. *ITS1* and *ITS2* sequences showed much lower intraspecific variations than *COI* except in a few species/populations (S4 Table).

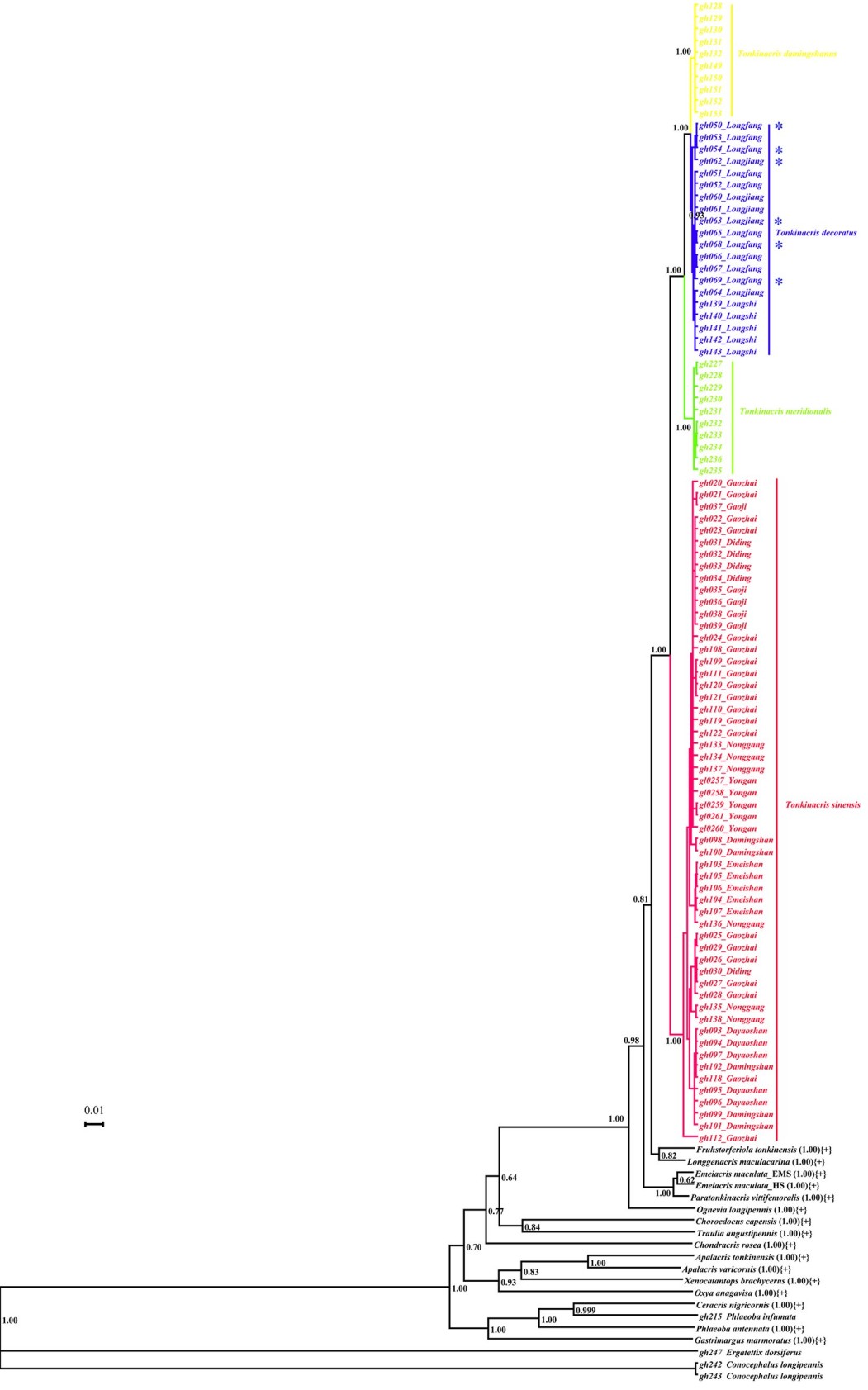

**Fig 4. BI tree deduced from combined sequences of mitochondrial *COI*, nuclear *ITS1* and *ITS2*.** Subclades of species with more than one individual sampled are collapsed and marked with {+} except those of the four species of *Tonkinacris*. The number in parentheses indicates its posterior probability value. The asterisk (*) indicates the individuals of *T. decoratus* with distinct or indistinct black transverse maculation on the base of the upper surface of hind femur.

Interspecific divergences of *COI* sequences within the genus *Tonkinacris* ranged from 1.02% (between *T. decoratus* and *T. damingshanus*) to 6.21% (between *T. decoratus* and *T. sinensis*), and those between other species pairs were all more than 4.76% (S5 Table). The interspecific divergences calculated from *ITS1* and *ITS2* sequences displayed similar distribution patterns to those from *COI* (S6 and S7 Tables). Species within *Tonkinacris* had an interspecific mean distance less than 1%, but the mean distances between other pairwise species within the subfamily Melanoplinae were usually more than 1%, and the mean distances between species in Melanoplinae and those out of Melanoplinae were all more than 10%.

## Species boundary delimitation of the genus *Tonkinacris*

To explore the species boundary within the genus *Tonkinacris*, we sampled 56 individuals for *T.sinensis* from 8 populations, 20 individuals for *T.decoratus*, 10 individuals for each species of *T. damingshanus* and *T. meridionalis* for comparison. Considering the poor performance of *ITS1* and *ITS2* sequences in resolving phylogeny, we used only *COI* sequences to estimate the gap between intra- and inter-specific distances within the genus *Tonkinacris*. The results showed that all species pairs, except *T. decoratus* and *T. damingshanus*, had distinct gaps between intraspecific variations and interspecific divergences calculated from *COI* sequences. Although there is no overlap between *T. decoratus* and *T. damingshanus*, the intra- and interspecific pairwise distances of them also had no gap (Table 2).

In NJ trees from both *COI* and combined sequences (S7 and S8 Figs), all the four species formed reciprocally monophyletic clades. While the clades of three species have a bootstrap value of 100, the one of *T. decoratus* is only 63 and 73, respectively, indicating the possibly instable relationship between *T. decoratus* and *T. damingshanus*. The 6 individuals of *T. decoratus* with distinct or indistinct black transverse maculation on the base of the upper surface of hind femur did not clustered into an independent clade either by themselves or with individuals of *T. damingshanus*. The relationship among *Tonkinacris* species was entirely unresolved in NJ trees from *ITS1*and *ITS2* sequences (S9 and S10 Figs).

For *COI* sequences, haplotype network analysis detected 17 haplotypes in *T. sinensis* (S8 Table), of which haplotype 1 is shared by four populations, haplotype 3, 8 and11 by two populations, respectively, and haplotype 6 by three populations (Table 3), indicating the close gene link among all sampled populations. In addition, the individuals of *T. decoratus* with and those without black transverse maculation on the base of the upper surface of hind femur can share

**Table 1. Mean genetic distances of *Tonkinacris sinensis* between populations.**

|           | Gaozhai | Diding | Gaoji | Dayaoshan | Damingshan | Emeishan | Nonggang |
|-----------|---------|--------|-------|-----------|------------|----------|----------|
| Diding    | 0.85%   |        |       |           |            |          |          |
| Gaoji     | 0.73%   | 0.40%  |       |           |            |          |          |
| Dayaoshan | 1.55%   | 1.53%  | 1.54% |           |            |          |          |
| Damingshan| 1.46%   | 1.36%  | 1.26% | 0.80%     |            |          |          |
| Emeishan  | 1.47%   | 1.29%  | 1.14% | 1.70%     | 1.54%      |          |          |
| Nonggang  | 1.09%   | 0.89%  | 0.72% | 1.54%     | 1.40%      | 1.13%    |          |
| Yong'an   | 0.77%   | 0.46%  | 0.18% | 1.67%     | 1.39%      | 1.26%    | 0.85%    |

**Table 2. Intraspecific variation of and interspecific divergence between species of the genus *Tonkinacris* calculated from *COI* sequences.**

| Species | Intraspecific variation (Pairwise/mean) | Interspecific divergence (Pairwise/mean) | | |
|---|---|---|---|---|
| | | *T. sinensis* | *T. decoratus* | *T. damingshanus* |
| *T. sinensis* | 0–3.29% (1.10%) | | | |
| *T. decoratus* | 0–0.92% (0. 31%) | 5.73–7.07% (6.21%) | | |
| *T. damingshanus* | 0–0.15% (0. 03%) | 6.06–6.57% (6.18%) | 0.92–1.38% (1.02%) | |
| *T. meridionlis* | 0–0.30% (0. 26%) | 5.07–6.08% (5.70%) | 2.64–3.12% (2.73%) | 2.96–3.13% (2.98%) |

the same haplotype (S8 Table). Haplotypes 16 and 17 are private for Yong'an population, but Yong'an and Gaozhai are located at the west and east sides of Maoershan Nature Reserve, respectively and can be regarded as the same locality in a larger scale. Five haplotypes were detected in *T. decoratus* and 6 were discovered in *T. meridionalis*. However, only 2 haplotypes were detected in *T. damingshanus*. For *ITS1* sequences, the number of haplotypes detected in the four species are 11, 3, 2 and 2, respectively, and no haplotype was shared by different species (S9 Table). For *ITS2* sequences, only 8 haplotypes were detected in the genus *Tonkinacris*, of which haplotypes 1, 2, 3, 5 and 6 were private for *T. sinensis*, haplotype 8 was private for *T. damingshanus*, haplotype 4 was shared by *T. sinensis* and *T.decoratus*, and haplotype 7 was shared by all of the four *Tonkinacris* species (S10 Table).

In the network from *COI* haplotypes (Fig 5A), *T. sinensis* formed two separate clades with one consisting of the single gh112 only, and the other one comprising the 16 remaining haplotypes. *T. meridionalis* formed an independent clade, but *T. decoratus* and *T. damingshanus* clustered into the same clade together. While the two haplotypes of *T. damingshanus* formed a separate subclade, the mutation steps between *T. decoratus* and *T. damingshanus* were 6, distinctly less than the maximum connection steps of 11 at 95% connection limit. In the network from *ITS1* haplotypes (Fig 5B), all of the four *Tonkinacris* species clustered into a single network, but haplotypes of each species formed a separate subclade. In the network from *ITS2* haplotypes (Fig 5C), the four species not only formed a single network together, but also had two shared haplotypes, with one shared by *T. sinensis* and *T. decoratus*, and the other one shared by all of the four *Tonkinacris* species.

In GMYC analysis based on *COI* sequences, 26 putative species were delineated from the whole dataset (S11 Table). The results of GMYC analysis for *Paratonkinacris vittifemoralis* and *Emeiacris maculata* were the same as those in the previous study [22]. *T. decoratus* and *T. damingshanus* were delineated as the same species. Samples of *T. sinensis* were delineated into 4 putative species, and samples of each remaining species were delineated as an independent species. For the 4 putative species delineated from samples of *T. sinensis* in GMYC analysis (S11 Table), the putative species 5 consisted of individuals from 3 populations (Dayaoshan, Damingshan and Gaozhai), the putative species 6 consisted of individuals from 7 populations (Gaozhai, Yong'an, Gaoji, Damingshan, Diding, Nonggang, Emeishan), the putative species 7

**Table 3. Haplotypes of *COI* sequences shared by populations of *T.sinensis*.**

| Population | Gaozhai | Gaoji | Diding | Nonggang | Dayaoshan | Damingshan | Emeishan |
|---|---|---|---|---|---|---|---|
| Haplotype 1 | √ | √ | √ | √ | | | |
| Haplotype 3 | √ | | √ | | | | |
| Haplotype 6 | √ | | | | √ | √ | |
| Haplotype 8 | | | | | √ | √ | |
| Haplotype 11 | | | | √ | | | √ |

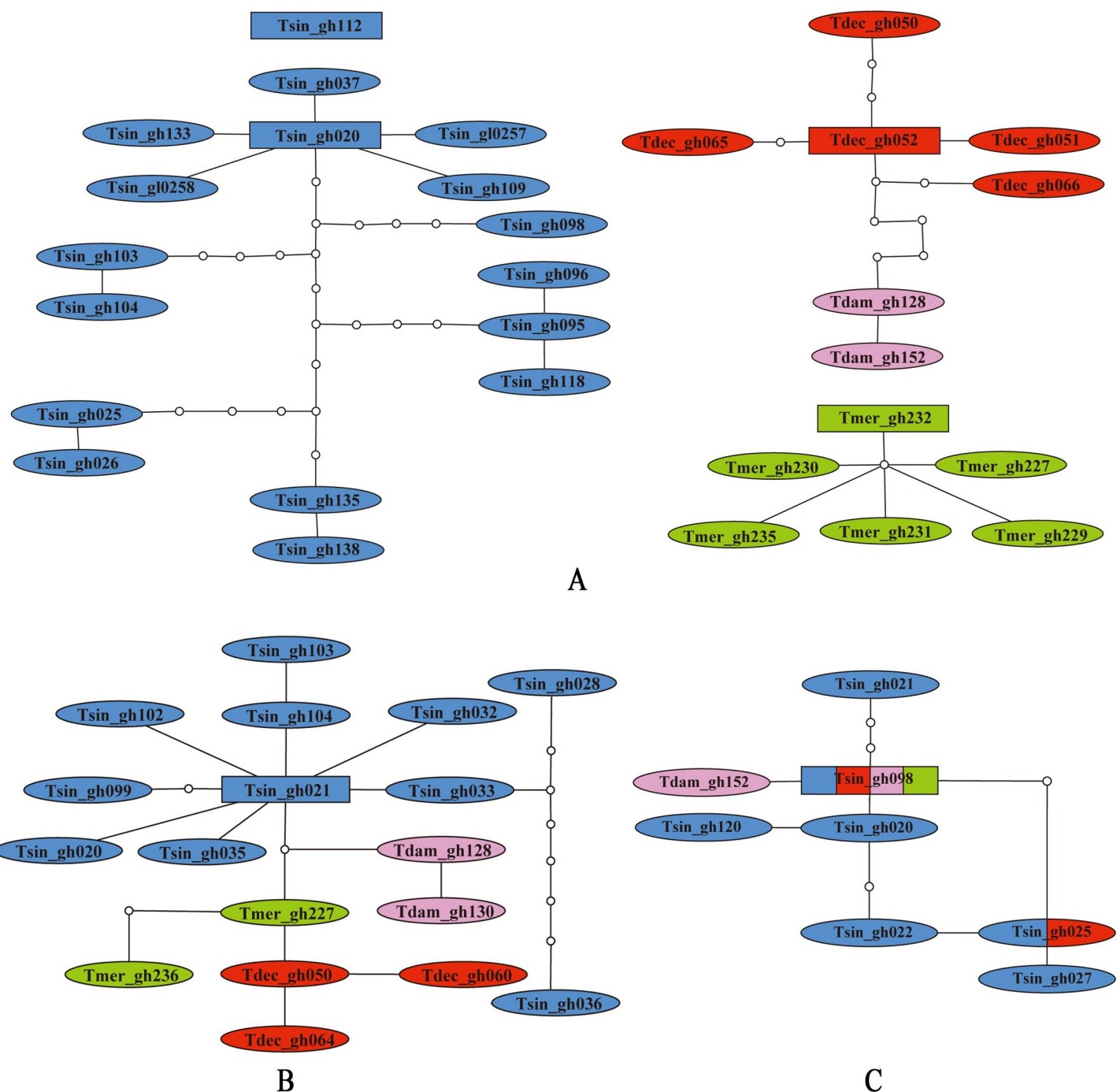

**Fig 5. Haplotype networks of *Tonkinacris spp*.** A. Networks reconstructed from *COI* sequences. B. Network reconstructed from *ITS1* sequences. C. Network reconstructed from *ITS2* sequences.

consisted of individuals from 3 populations (Gaozhai, Diding, Nonggang), and the putative species 8 consisted of the single individual gh112 from Gaozhai population. Samples of Gaozhai population were delineated into 4 putative species, and samples of Damingshan, Diding and Nonggang populations were delineated into 2 putative species, respectively.

## Discussion

### Phylogeny of the genus *Tonkinacris*

The molecular marker of *COI* has been demonstrated informative and useful for comparison within and between species in Acrididae in some previous studies [18,22,43–45]. While nuclear *ITS1* and *ITS2* sequences could not resolve the phylogeny within the genus *Tonkinacris* by any approach of ML, BI and NJ analyses (S1, S2, S5, S6, S9 and S10 Figs), mitochondrial *COI* barcode sequences performed an excellent resolution. The monophyly of the genus *Tonkinacris* and the four species of the genus had been retrieved in most phylogenetic trees only if the *COI* sequences were used (Figs 3 and 4; S5, S8 and S9 Figs), except in the ML tree from combined sequences where the monophyly of the genus *Tonkinacris* and the species *T. sinensis* was not supported because of the serious effects of *ITS1* and *ITS2* sequences (S3 Fig). Similarly, the relationship among species within the genus *Tonkinacris* was consistently resolved as (*T. sinensis*, (*T. meridionalis*, (*T. decoratus*, *T. damingshanus*))) (Figs 3 and 4; S5, S8 and S9 Figs) even in the ML tree from combined sequences where the monophyly of each of the three species, *T. meridionalis*, *T. decoratus* and *T. damingshanus*, was still retrieved despite of the poor resolution of *ITS1* and *ITS2* sequences (S3 Fig). Since there is possibility that *ITS1* and *ITS2* sequences are not suitable for addressing phylogenetic problems below subfamily level because of their high homology [22,46], and the morphology agrees to a large extent with the results from molecular datasets comprising either single *COI* or combined sequences, it is reasonable to provisionally propose such a hypothesis that the Chinese species of the genus *Tonkinacris* is a monophyletic group and the phylogenetic relationship can be describedas (*T. sinensis*, (*T. meridionalis*, (*T. decoratus*, *T. damingshanus*))).

Although *T. ruficerus* and *T. yaeyamaensis* distributed in Japan were not involved in this study, we believe that it does not matter because they may not belong to the genus *Tonkinacris* due to their distinct morphological differences from their congeners in continental China and Vietnam just as mentioned in the introduction section.

### Species delimitation within the genus *Tonkinacris*

There is no doubt that both *T. sinensis* and *T. meridionalis* are good independent species because: (1) they are not only distinguishable morphologically (Fig 2) but also supported by molecular evidence, i.e. the monophyly of each species is supported in nearly all phylogenetic trees from *COI* and combined datasets with extremely high bootstrap values or posterior probabilities (Figs 3 and 4; S5, S8 and S9 Figs) except in the ML tree from combined dataset (S3 Fig); (2) there are distinct gaps between their intra- and inter-specific genetic distances (Table 2), and (3) haplotypes of *COI* of each species formed a separate network (Fig 5A) except the haplotype represented by gh112, which is the only extreme sequence and possibly derived from sequencing error. However, the relationship between *T. decoratus* and *T. damingshanus* is not so unambiguous. While the monophyly of the two species was supported by *COI* and combined datasets most of the time, there was no gap for them between intra- and inter-specific distances (Table 2). The differences in morphological character between *T. decoratus* and *T. damingshanus* proposed in original reference was not supported by molecular evidence in this study, but improved to be a natural variation. Furthermore, the mutation steps of 6 between them in the network from *COI* sequences is distinctly less than the maximum connection steps of 11 at 95% connection limit. More importantly, there are only two haplotypes for *T. damingshanus* detected from 10 samples, whereas 5 haplotypes were detected for *T. meridionalis* from the same amount of samples. Despite of the similar ratio of haplotypes detected for *T. decoratus*, the increased number of samples led to an increase of haplotype amount detected

(S8 Table). Therefore, it is possible that conspicuous overlap for *T. decoratus* and *T. damingshanus* between intra- and inter-specific distances will arise when more individuals are sampled. Finally, *T. decoratus* and *T. damingshanus* were delineated as the same species in GMYC analysis. Since the only distinguishing morphological character between *T. decoratus* and *T. damingshanus* disappears and they were delineated as the same species by molecular approaches most of the time, it is appropriate to treat them as the same species, and a formal nomenclatural change will be made when the genus is revised based on the integrative evidences in the near future.

As for the 4 putative species delineated from samples of *T. sinensis* in GMYC analysis, they can be interpreted as a result of oversplitting because one of the four GMYC species (putative species 6) covers individuals from seven of the eight populations sampled (S11 Table), and this splitting was not supported by either morphological evidence or results from other molecular approaches, including the monophyly in most of the phylogenetic trees, the intraspecific genetic distances less than 2.00% most of the time, the close relationships among populations linked through shared haplotypes, and the single independent haplotype network (excluding the extreme one, gh112).

## Performance of *ITS1* and *ITS2* sequences in resolving phylogeny of grasshoppers

Sequences of ribosomal DNA (rDNA), including both ribosomal RNA (rRNA) genes and their associated spacer regions, may be used to address phylogenetic problems at a variety of taxonomic levels [47]. Coding sequences are fairly conservative and have been used for examining relationships among more distant taxa. In contrast, spacer regions, which bracket the large and small coding elements, are subject to fewer constraints and, consequently, evolve more rapidly, supposedly making them useful for deducing relationships among strains or closely related groups [48–50]. However, this may not be the case in all groups of organisms. While the sequences of *ITS1* in the genus *Melanoplus* had appreciable differences when insertions and deletions were taken into account, its utility might be questionable because of the long stretches of very high homology in this region [46]. Despite the fact that *ITS2* sequences have been used to explore phylogeny and delimitate species boundary in some grasshopper groups [51–53], our previous [22] and present studies achieved similar results to the supposition by Kuperus & Chapco [46]. In this study, the relationship within and between closely species was nearly completely unresolved using *ITS1* and *ITS2*, but the relationship at the subfamily level was resolved much better, and the monophyly of the subfamily Melanoplinae and some distantly related species was retrieved with high bootstrap values or posterior probabilities in most phylogenetic trees from *ITS1* or *ITS2* (S2, S5, S6, S9 and S10 Figs), except in the ML tree from *ITS1* (S1 Fig) in which the monophyly of the subfamily Melanoplinae was not supported. Therefore, *ITS1* and *ITS2* sequences may not be suitable for addressing phylogenetic problems in Acrididae at genus and species levels, but may be usable at or above subfamily levels.

## Supporting information

**S1 Fig. ML tree deduced from ITS1 sequences.**
(DOCX)

**S2 Fig. ML tree deduced from ITS2 sequences.**
(DOCX)

**S3 Fig. ML tree deduced from combined sequences of mitochondrial *COI*, nuclear ITS1 and ITS2.**
(DOCX)

**S4 Fig. BI tree deduced from COI sequences.**
(DOCX)

**S5 Fig. BI tree deduced from ITS1 sequences.**
(DOCX)

**S6 Fig. BI tree deduced from ITS2 sequences.**
(DOCX)

**S7 Fig. NJ tree deduced from COI sequences.**
(DOCX)

**S8 Fig. NJ tree deduced from combined sequences of mitochondrial *COI*, nuclear ITS1 and ITS2.**
(DOCX)

**S9 Fig. NJ tree deduced from ITS1 sequences.**
(DOCX)

**S10 Fig. NJ tree deduced from ITS2 sequences.**
(DOCX)

**S1 Table. Molecular data of the genus *Tonkinacris* submitted to NCBI so far.**
(DOCX)

**S2 Table. Materials involved in this study.**
(DOCX)

**S3 Table. Mapping table between GenBank accession numbers and voucher numbers.**
(DOCX)

**S4 Table. Intraspecific variations calculated from different datasets.**
(DOCX)

**S5 Table. Mean genetic distances between species calculated from *COI* alignment.**
(DOCX)

**S6 Table. Mean genetic distances between species calculated from ITS1 alignment.**
(DOCX)

**S7 Table. Mean genetic distances between species calculated from ITS2 alignment.**
(DOCX)

**S8 Table. Haplotyptes of COI detected from samples of *Tonkinacris spp*.**
(DOCX)

**S9 Table. Haplotyptes of ITS1 detected from samples of *Tonkinacris spp*.**
(DOCX)

**S10 Table. Haplotyptes of ITS2 detected from samples of *Tonkinacris spp*.**
(DOCX)

**S11 Table. Putative species delineated from COI alignment using GMYC model.**
(DOCX)

## Acknowledgments

We would like to thank Mr. Tao Wang for his help in implementing GMYC analysis, Dr. Zhi-jun Zhou for his instructions on processing the molecular data. Special thanks will be given to Mr. Ruigang Yang, Chunwen Lu and Bingchui Su for collecting partial materials for molecular study.

## Author Contributions

**Conceptualization:** Yuan Huang, Jianhua Huang.

**Data curation:** Jingxiao Gu, Tao Wei, Liliang Lin, Jianhua Huang.

**Formal analysis:** Haojie Wang, Bing Jiang, Jingxiao Gu, Tao Wei, Liliang Lin.

**Funding acquisition:** Liliang Lin, Dan Liang, Jianhua Huang.

**Investigation:** Tao Wei, Liliang Lin.

**Methodology:** Yuan Huang.

**Project administration:** Jianhua Huang.

**Supervision:** Yuan Huang, Dan Liang, Jianhua Huang.

**Visualization:** Bing Jiang, Jingxiao Gu, Jianhua Huang.

**Writing – original draft:** Haojie Wang, Bing Jiang.

**Writing – review & editing:** Yuan Huang, Dan Liang, Jianhua Huang.

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
