## [Decision Letter · Decision Letter 0]

4 Mar 2021

PONE-D-20-39875

Molecular phylogeny and species delimitation of the genus Tonkinacris (Orthoptera, Acrididae, Melanoplinae) from China

PLOS ONE

Dear Dr. Huang,

Thank you for submitting your manuscript to PLOS ONE. After careful consideration, we feel that it has merit but does not fully meet PLOS ONE’s publication criteria as it currently stands. Therefore, we invite you to submit a revised version of the manuscript that addresses the points raised during the review process.

We look forward to receiving your revised manuscript.

Kind regards,

Feng ZHANG, Ph.D.

Academic Editor

PLOS ONE

Journal Requirements:

3. We note that Figure 1 in your submission contain map images which may be copyrighted. All PLOS content is published under the Creative Commons Attribution License (CC BY 4.0), which means that the manuscript, images, and Supporting Information files will be freely available online, and any third party is permitted to access, download, copy, distribute, and use these materials in any way, even commercially, with proper attribution. For these reasons, we cannot publish previously copyrighted maps or satellite images created using proprietary data, such as Google software (Google Maps, Street View, and Earth). For more information, see our copyright guidelines: http://journals.plos.org/plosone/s/licenses-and-copyright.

3.1.    You may seek permission from the original copyright holder of Figure 1 to publish the content specifically under the CC BY 4.0 license. 

3.2.    If you are unable to obtain permission from the original copyright holder to publish these figures under the CC BY 4.0 license or if the copyright holder’s requirements are incompatible with the CC BY 4.0 license, please either i) remove the figure or ii) supply a replacement figure that complies with the CC BY 4.0 license. Please check copyright information on all replacement figures and update the figure caption with source information. If applicable, please specify in the figure caption text when a figure is similar but not identical to the original image and is therefore for illustrative purposes only.

Reviewers' comments:

Reviewer's Responses to Questions

**Comments to the Author**

1. Is the manuscript technically sound, and do the data support the conclusions?

Reviewer #1: Yes

Reviewer #2: Yes

2. Has the statistical analysis been performed appropriately and rigorously? 

Reviewer #1: Yes

Reviewer #2: Yes

3. Have the authors made all data underlying the findings in their manuscript fully available?

Reviewer #1: Yes

Reviewer #2: Yes

4. Is the manuscript presented in an intelligible fashion and written in standard English?

Reviewer #1: Yes

Reviewer #2: No

5. Review Comments to the Author

Reviewer #1: Wang et al. performed a molecular phylogeny and species delimitation of the genus Tonkinacris (Orthoptera, Acrididae, Melanoplinae) from China. I do think it’s an interesting study and is acceptable for publication in PLOS ONE. I just list below a few tiny stuffs which are needed to be double checking by the authors.

1. For the study of molecular phylogeny with the genus Tonkinacris, the sample from another suborder/family is totally superfluous.

2. Line 56: “phylogenetic relationship within the genus has never been explored until now.” change to “phylogenetic relationship within the genus Tonkinacris has never been explored until now.”

3. Line 57: “As for species identification of the genus,…”change to “For morphological species identification, …”

4. Line 82-83: “The two Japanese Tonkinacris species were excluded from this study…”change to “The two Japanese Tonkinacris species were not included in this study…”

5. Line 152: “with 10 million MCMC generations” change to “with 1×107 MCMC generations”

6. Line 153: “a burn-in of one million generations” change to “a burn-in of the first 10% generations”

7. Line 177: “Phylogeny revealed by BI analysis was similar to that by ML analysis.” change to “Phylogeny revealed by BI analyses was similar to that by ML analyses.”

8. Line 203: “less or slightly more than 1%” change to “less or slightly above 1%”

9. Line 327: “Although the two species of the genus Tonkinacris distributed in Japan…” change to “Although T. ruficerus and T. yaeyamaensis distributed in Japan…”

Reviewer #2: Although Tonkinaccris is a small genus with six species in the world, while only 4 species in China, this manuscript provides a detailed discussion of the genus in terms of morphological and molecular information. Therefore, I think this manuscript may be considered for publication. However, there are the following problems.

1) On the page 4,lines 162-165, “However, the monophyly for the genus Tonkinacris as well as the sepcies T. sinensis, T. meridionalis, Longgenacris maculacarina, Paratonkinacris vittifemoralis and Emeiacris maculata was not supported in the ML tree from ITS1 sequences”, I cannot understand. Since the sepcies T. sinensis and T. meridionalis belong to the genus Tonkinacris, why to use “as well as” ?

2) On the page 8, lines 343-346, the linguistic expression is problematic, because morphological character and molecular evidence are two distinct concepts. They can say that the differences in morphological character are not supported by molecular differences.

3) All figures are not clear. Low photo resolution.

4) Chinglish is more prominent in this manuscript. English speakers are requested to polish this draft.

6. PLOS authors have the option to publish the peer review history of their article (what does this mean?). If published, this will include your full peer review and any attached files.

Reviewer #1: No

Reviewer #2: No

---

## [Author Response · Author response to Decision Letter 0]

13 Mar 2021

1. We have fully reviewed the reference list to ensure that it is complete and correct.

 2. We have formatted the manuscript to meet PLOS ONE's style requirements according to the guidlines in the PLOS ONE style templates.

 3. We have no amended statement on financial disclosure for the manuscript. 

 4. The ground map of figure 1 in the manuscript meets the requirement of publication in PLOSOne. There is no need to get a written permission from the copyright holder.

 5. We have revised the main text according to the comments of the reviewers

 6. All of the figures uploeaded onto the system have enough resolutions.

 7. We have invite native English speakers to help us polish the manuscript.

---

## [Editor Report · Decision Letter 1]

18 Mar 2021

Molecular phylogeny and species delimitation of the genus Tonkinacris (Orthoptera, Acrididae, Melanoplinae) from China

PONE-D-20-39875R1

Dear Dr. Huang,

We’re pleased to inform you that your manuscript has been judged scientifically suitable for publication and will be formally accepted for publication once it meets all outstanding technical requirements.

Kind regards,

Feng ZHANG, Ph.D.

Academic Editor

PLOS ONE
---

## [Editor Report · Acceptance letter]

29 Mar 2021

PONE-D-20-39875R1 

Molecular phylogeny and species delimitation of the genus *Tonkinacris* (Orthoptera, Acrididae, Melanoplinae) from China 

Dear Dr. Huang:

I'm pleased to inform you that your manuscript has been deemed suitable for publication in PLOS ONE. Congratulations! Your manuscript is now with our production department. 

Kind regards, 

on behalf of

Dr. Feng ZHANG 

Academic Editor

PLOS ONE